# Change in five-factor model personality traits during the acute phase of the coronavirus pandemic

**Angelina R. Sutin**[1]*, **Martina Luchetti**[1], **Damaris Aschwanden**[2], **Ji Hyun Lee**[1], **Amanda A. Sesker**[1], **Jason E. Strickhouser**[1], **Yannick Stephan**[3], **Antonio Terracciano**[2]

**1** Department of Behavioral Sciences and Social Medicine, Florida State University College of Medicine, Tallahassee, Florida, United States of America, **2** Department of Geriatrics, Florida State University College of Medicine, Tallahassee, Florida, United States of America, **3** Euromov, University of Montpellier, Montpellier, France

* angelina.sutin@med.fsu.edu

**Data Availability Statement:** All relevant data are deposited at DOI:10.17605/OSF.IO/5AF3U.

**Funding:** Research reported in this publication was supported by the National Institute on Aging of the National Institutes of Health under Award Number

## Abstract

The rapid spread of the coronavirus and the strategies to slow it have disrupted just about every aspect of our lives. Such disruption may be reflected in changes in psychological function. The present study used a pre-posttest design to test whether Five Factor Model personality traits changed with the coronavirus outbreak in the United States. Participants ($N = $ 2,137) were tested in early February 2020 and again during the President's 15 Days to Slow the Spread guidelines. In contrast to the preregistered hypotheses, Neuroticism decreased across these six weeks, particularly the facets of Anxiety and Depression, and Conscientiousness did not change. Interestingly, there was some evidence that the rapid changes in the social context had changed the meaning of an item. Specifically, an item about going to work despite being sick was a good indicator of conscientiousness before COVID-19, but the interpretation of it changed with the pandemic. In sum, the unexpected small decline in Neuroticism suggests that, during the acute phase of the coronavirus outbreak, feelings of anxiety and distress may be attributed more to the pandemic than to one's personality.

## Introduction

The world is facing an extraordinary crisis. The novel coronavirus that emerged and started to spread at the end of 2019 is now reported in nearly every country and territory in the world [1]. The response to control the spread has been equally extraordinary. During the acute phase of the pandemic, entire countries were put on lock down to slow the spread [2]. The United States restricted international travel [3] and most states issued varying degrees of "stay at home" orders [4]. Such measures are essential to slow the spread of the virus [5]. The concerns over the virus and the stress associated with the social restrictions may have acute psychological consequences [6].

Here we test whether there are acute changes in Five Factor Model (FFM; [7]) personality traits and facets in response to the emerging coronavirus pandemic. FFM traits are stable

R01AG053297 to ARS. The content is solely the responsibility of the authors and does not necessarily represent the official views of the National Institutes of Health. The funders had no role in study design, data collection and analysis, decision to publish, or preparation of the manuscript.

**Competing interests:** The authors have declared that no competing interests exist.

individual differences that tend to be resistant to normative stressful life events [8]. Although typically stable, evidence from the psychopathology literature [9] and intervention research [10] indicates that traits can and do change in response to distress and treatment for distress, respectively. The coronavirus outbreak and measures to control its spread have disrupted most aspects of life, including basic motives (e.g., relationships, work) and daily activities that have been fundamental to work on adult personality development [11].

Using a sample that was first assessed in late January and early February 2020 and then again in mid-March 2020 during the President's 15 Days to Slow the Spread guidelines [12], we tested for acute personality change in response to the coronavirus pandemic. Under normal circumstances, there is no reason to expect that personality would change over such a short period of time. Given the extraordinary nature of the coronavirus pandemic, and the drastic measures that have been taken to control its spread, however, personality may be reactive to these rapidly changing events. We tested the preregistered hypothesis that Neuroticism, particularly the anxiety facet of this trait, increased between pre- and post-test because the collective worry and anxiety over the virus would increase a trait tendency toward worry and anxiety. We also hypothesized that Conscientiousness, particularly facets related to responsibility, would increase between pre- and post-test because the public health messaging on the importance of personal responsibility to control the spread (e.g., handwashing, social distancing) would consolidate into a greater tendency toward rule-following and responsibility (i.e., conscientiousness), particularly in relation to others. We did not make directional hypotheses for the other three traits. Exploratory analyses examined whether there were larger changes in adults older than 65, males, and those in isolation/quarantine status because of the greater risk in these groups.

## Materials and methods

### Participants and procedure

The present study used a pre-post design with all participants exposed to the stressor (coronavirus pandemic). Preregistration for this study can be found at https://osf.io/vqnh8/?view_only=8660007bc8ef4f168e07af15f9c49f43. The Institutional Review Board at the Florida State University approved this study (STUDY00000003). Prior to both surveys, potential participants were given a brief description of the survey including the content area of the questionnaires. To continue with the questionnaire, individuals had to indicate that they understood the survey would include health-related questions, that they were 18 years or older, and that they wanted to participate in the study. Individuals who clicked yes were directed to the survey; individuals who clicked no were routed out of the survey. Documentation of informed consent was waived because the data were collected and analyzed anonymously from web-based surveys.

**Pretest.** An online survey was fielded between January 31 and February 10, 2020 (not preregistered). At this time, the coronavirus was spreading in Asia but had not spread yet in the United States. The purpose of this original survey was to examine the physical, social, and cognitive correlates of FFM personality traits and well-being across adulthood. We contracted with Dynata (www.dynata.com) to recruit participants to complete a Qualtrics survey administered by the Florida State University College of Medicine. Participants were sampled from across the United States and stratified in equal numbers ($n$ = 500) across seven age bands: 18–19, 20–29, 30–39, 40–49, 50–59, 60–69, and 70 and older. The adolescent age band was smaller than the other 10-year age bands because it was restricted to participants who could consent legally to participate without consent from a parent or legal guardian. The sample was also stratified by gender (50%/50% male/female) and race (20% African American). The stratification did not go as intended in the original data collection and there was an oversample

of participants between 30–59 years old and few participants who were 18 or 19. More participants in the 18–19 year-old age band were recruited and tested to improve representation in this age group. As such, in the final sample, participants between 30 and 59 years old were oversampled relative to the other age groups, and the overall sample size was larger than originally planned. A total of 3,963 participants had valid personality data at pretest (See S1 Fig for a flowchart of participant inclusion at pretest).

**Posttest.** A second online survey was fielded between March 18 and March 29, 2020 (preregistered). Participants were again recruited through Dynata. All participants who completed the personality measure at pretest were invited to complete the posttest survey. The exception was participants who had left the Dynata panel in between pretest and posttest. Dynata does not recontact people who have left their panels, and we could not contact them directly because we had no personally identifying or contact information for any participant. When participants clicked on the link from Dynata, the description of the survey was similar to the description of the first survey, except that participants were told that they would also be asked questions about current events. Participants were also told that the survey would include some questions that were similar to what they might have been asked before and that it was important to answer the questions, even if the questions had been answered before. The personality measure was embedded in a larger survey block that included questions about other aspects of psychological functioning, health, and health-related behavior (see preregistration for full questionnaire). The survey also included a block of questionnaires about COVID-19. These two blocks were randomly counterbalanced across participants. There was no effect of order on any of the results reported below.

A total of 2,137 participants had valid data on personality traits at both pretest and posttest (See S1 Fig for a flowchart of participant inclusion at posttest). Participants in the final analytic sample were from all 50 states and Washington, DC and Puerto Rico and participated in numbers roughly proportional to the population of the state/DC/PR (i.e., sample sizes were larger in populous states, such as California and New York, and smaller in less populous states, such as North Dakota and Wyoming).

Attrition analyses indicated that compared to those with posttest data, participants who did not have posttest data were younger ($d$ = .92, $p$ = .000), more likely to be female ($\chi^2$ = 66.754, $p$ = .000), more likely to be African American ($\chi^2$ = 45.345, $p$ = .000), more likely to be Latinx ethnicity ($\chi^2$ = 93.122, $p$ = .000), had less education ($d$ = .38, $p$ = .000), scored higher in Neuroticism ($d$ = .41, $p$ = .000), higher in Openness ($d$ = .07, $p$ = .031), lower in Agreeableness ($d$ = .26, $p$ = .000) and Conscientiousness ($d$ = .45, $p$ = .000); there was no difference in Extraversion ($d$ = .02, $p$ = .520). Accounting for age and sex differences in attrition, the effect size for the difference by attrition was reduced for Neuroticism ($d$ = .09, $p$ = .007), Conscientiousness ($d$ = .12, $p$ = .000) and Agreeableness ($d$ = .01, $p$ = .723); there was a larger difference in Openness ($d$ = .13, $p$ = .000).

## Measures

**Personality traits.** FFM personality traits were measured with the Big Five Inventory-2 (BFI-2; [13]), a 60-item measure of the five broad domains and three more circumscribed facets per domain. Items completed the sentence stem, "I am someone who. . ." and measured Neuroticism (e.g., worries a lot), Extraversion (e.g., is outgoing, sociable), Openness (e.g., is complex, a deep thinker), Agreeableness (e.g., is compassionate, has a soft heart), and Conscientiousness (e.g., is systematic, likes keeping things in order). Responses were made on a scale from 1 (*strongly disagree*) to 5 (*strongly agree*). Items were reverse scored in the direction of the trait label when necessary and the mean taken across items. In addition to the five broad

domains, three facets for each trait were scored: Anxiety, Depression, and Emotional Volatility for Neuroticism, Sociability, Assertiveness, and Energy Level for Extraversion, Intellectual Curiosity, Aesthetic Sensitivity, and Creative Imagination for Openness, Compassion, Respectfulness, and Trust for Agreeableness, and Organization, Productiveness, and Responsibility for Conscientiousness. The same measure was administered at both assessments.

Two additional facets of Conscientiousness were measured at both assessments: The Responsibility facet from Roberts and colleagues' facet measure of Conscientiousness [14], and the Dutifulness facet from the NEO-PI-3 [15]. The Responsibility facet was measured with four items (e.g., "I go out of my way to keep my promises."). The Dutifulness facet was measured with eight items (e.g., "When I make a commitment, I can always be counted on to follow through."). All items were rated on a scale from 1 (*strongly disagree*) to 5 (*strongly agree*). Items were reverse scored in the direction of the trait label when necessary and the mean taken across items. The same measures were administered at both assessments.

**Quarantine/Isolation status.** At posttest, participants responded (no/yes) to the item "In the last month, I have been in quarantine/isolation because of the coronavirus." This item did not differentiate between quarantine for medical reasons and voluntary isolation, and thus it was not possible to disentangle the potential effects of the purpose of the quarantine/isolation.

**Covariates.** Participants reported their age in years, gender identification (male, female, transgender, other/unknown), race, ethnicity, and education (from 1 = less than high school to 7 = PhD or equivalent). Gender was coded to compare females (= 1) to males (= 0). Participants who identified as transgender/other/unknown (*n* = 11) were included with the female category (results did not vary if these participants were not included in the analysis). Race was coded as African American/black (= 1) compared to all others (= 0). Ethnicity was coded as Latinx/Hispanic (= 1) compared to all others (= 0). Participants of other races (e.g., Asian) were categorized with the "all others" group. In some analyses, age was coded into older adults (65 and older; = 1) versus younger and middle-aged adults (18–64; = 0) because coronavirus poses a greater threat to older than younger adults [16].

## Analytic strategy

Repeated measures analysis of variance (ANOVA) was used to test for mean trait change between pre- and post-test. This analysis was run for each trait and facet. We then used Repeated Measures Analysis of Covariance (ANCOVA) to test whether trait change varied by age group and gender (exploratory analyses in the preregistration). In additional exploratory analyses (not preregistered), we tested whether personality change varied by quarantine/isolation status. With a repeated measures design and a sample of 2,137 participants, we had >90% power to detect a small change (*d* = .1) at alpha < .05 (two-tailed).

## Results

Descriptive statistics are shown in Table 1. The test-retest correlation was high for all five traits (≥.80), supporting the reliability of the measure and data quality. We report the results by personality domain, starting with the two domains that were hypothesized to change. Note that the results of the preregistered analyses are presented in Table 2, the results of the preregistered exploratory analyses for age and gender are presented in S2 and S3 Tables, respectively, and the results of the non-preregistered exploratory analysis for isolation are presented in Table 3.

## Neuroticism

The repeated measures ANOVA indicated change in Neuroticism in the overall sample (Table 2): Compared to before the spread of the coronavirus, Neuroticism decreased (*d* = -.04) during the

**Table 1. Demographic characteristics of the sample.**

| Demographic Factor | Mean (SD) or % (*n*) |
|---|---|
| Age (years) | 51.022 (16.608) |
| Gender | |
| Male | 51.1% (1091) |
| Female | 47.9% (1024) |
| Other/unknown | 1.1% (22) |
| Race | |
| African American or Black | 16.9% (361) |
| Not African American | 83.1% (1776) |
| Ethnicity | |
| Latinx or Hispanic | 10.7% (229) |
| Not Latinx | 89.3% (1908) |
| Education[a] | 4.169 (1.517) |
| Quarantine/Isolation[b] | |
| Yes | 24.9% (524) |
| No | 75.1% (1582) |

*N* = 2,137.

[a] Reported on a scale from 1 (less than high school) to 7 (PhD or equivalent).

[b] *n* = 2,106 due to missing data.

acute phase of the pandemic in the United States. At the facet level, the Anxiety and Depression facets of Neuroticism decreased, whereas there was no change in Emotional Volatility. The decrease in overall Neuroticism and Anxiety remained significant controlling for the covariates (S1 Table). Age and gender did not moderate change in overall Neuroticism or any of the three facets (S2 and S3 Tables). The results do not support our hypothesis that domain-level Neuroticism and the facet of Anxiety would increase during the acute phase of the coronavirus pandemic in the United States. The effect found was opposite of what was expected, and the magnitude of change was small.

Approximately 25% of the sample reported being in quarantine/isolation in the last month. There were baseline differences in Neuroticism (*d* = .32) by subsequent isolation status and evidence that change in Neuroticism was moderated by isolation status (Table 3). Specifically, the decrease in Neuroticism was only apparent among participants not in isolation (d = .06); there was a slight non-significant increase for those in isolation (d = .01). A stronger cross-over effect was found for trait Depression. Specifically, participants not in isolation decreased (d = .08) in a tendency toward Depression whereas those in isolation increased (d = .06). Isolation did not moderate change in Anxiety or Emotional Volatility.

## Conscientiousness

The repeated measures ANOVA indicated no change in Conscientiousness across the two measurements (Table 2; *d* = .00). Although there was no change in the overall domain of Conscientiousness, two facets did change: The BFI-2 facet of Productiveness increased between pre- and post-test, whereas the NEO facet of Dutifulness decreased. Responsibility measured either with the BFI-2 or Roberts and colleagues' measure did not change and neither did Organization. Moderation analysis indicated that the change in Dutifulness was stronger among participants younger than 65 years than participants older than 65 years (S2 Table). There was no other moderation by age. There was a modest interaction with gender, such that men

**Table 2. Mean change in personality traits between pretest and posttest.**

| Personality Trait | Pretest | | Posttest | | Time | P | $\eta^2$ |
|---|---|---|---|---|---|---|---|
| | Mean | SD | Mean | SD | | | |
| Neuroticism | 2.611 | .779 | 2.576 | .800 | $F(1,2136) = 11.653$ | .001 | .005 |
| Extraversion | 3.120 | .641 | 3.138 | .645 | $F(1,2136) = 4.821$ | .028 | .002 |
| Openness | 3.454 | .624 | 3.465 | .639 | $F(1,2136) = 1.546$ | .214 | .001 |
| Agreeableness | 3.694 | .637 | 3.691 | .650 | $F(1,2136) = .083$ | .774 | .000 |
| Conscientiousness | 3.859 | .704 | 3.859 | .724 | $F(1,2136) = .002$ | .962 | .000 |
| Neuroticism Facets | | | | | | | |
| Anxiety | 2.912 | .895 | 2.864 | .890 | $F(1,2136) = 11.663$ | .001 | .005 |
| Depression | 2.430 | .899 | 2.394 | .917 | $F(1,2136) = 6.910$ | .009 | .003 |
| Emotional Volatility | 2.492 | .872 | 2.470 | .882 | $F(1,2136) = 2.539$ | .111 | .001 |
| Extraversion Facets | | | | | | | |
| Sociability | 2.926 | .900 | 2.908 | .902 | $F(1,2136) = 2.008$ | .157 | .001 |
| Assertiveness | 3.163 | .770 | 3.200 | .782 | $F(1,2136) = 8.645$ | .003 | .004 |
| Energy Level | 3.270 | .761 | 3.306 | .761 | $F(1,2136) = 7.848$ | .005 | .004 |
| Openness Facets | | | | | | | |
| Curiosity | 3.549 | .727 | 3.557 | .729 | $F(1,2136) = .514$ | .473 | .000 |
| Aesthetic Sensitivity | 3.286 | .831 | 3.300 | .839 | $F(1,2136) = 1.009$ | .296 | .000 |
| Creative Imagination | 3.527 | .768 | 3.537 | .781 | $F(1,2136) = .574$ | .449 | .000 |
| Agreeableness Facets | | | | | | | |
| Compassion | 3.770 | .772 | 3.777 | .776 | $F(1,2136) = .241$ | .623 | .000 |
| Respectfulness | 3.988 | .762 | 3.967 | .776 | $F(1,2136) = 3.281$ | .070 | .002 |
| Trust | 3.323 | .748 | 3.331 | .751 | $F(1,2136) = .384$ | .535 | .000 |
| Conscientiousness Facets | | | | | | | |
| Organization | 3.890 | .829 | 3.866 | .840 | $(1,2136) = 3.300$ | .069 | .002 |
| Productiveness | 3.793 | .808 | 3.820 | .813 | $(1,2136) = 5.073$ | .024 | .002 |
| Responsibility | 3.895 | .767 | 3.888 | .775 | $(1,2136) = .294$ | .588 | .000 |
| Responsibility[a]^ | 3.988 | .763 | 3.984 | .778 | $(1,2064) = .060$ | .807 | .000 |
| Dutifulness[a] | 3.897 | .610 | 3.859 | .603 | $(1,2024) = 12.037$ | .001 | .006 |

$N = 2{,}137$.

[a] Ns vary due to missing data.

^From Roberts et al. [14].

increased slightly and women/other genders decreased slightly in Conscientiousness; gender did not moderate change in the facets (S3 Table). Similar to Neuroticism, there were baseline differences in Conscientiousness by isolation status ($d = .38$) and an interaction, such that participants in isolation decreased slightly in Conscientiousness whereas participants not in isolation did not change, a change driven by somewhat larger changes in the facet of Organization (Table 3). Overall, the results do not support our hypothesis that domain-level Conscientiousness and the facet of Responsibility would increase during the acute phase of the coronavirus pandemic in the United States.

The Dutifulness scale included an item about the tendency to go to work or school even when not feeling well ("I try to go to work or school even when I'm not feeling well."). Given the public health messaging to stay at home if sick or even just possible exposure to someone who might have COVID-19, it was possible that the decline in overall Dutifulness was due to change on this item. To address this possibility, in exploratory analyses, we tested for change in this item and the other seven items on the Dutifulness scale (S4 Table). There was, in fact, a

**Table 3. Interaction between time and isolation on change in personality traits.**

| Personality Trait | Pre | | Post | | Time | Isolation | Time x Isolation |
|---|---|---|---|---|---|---|---|
| | Mean | SD | Mean | SD | | | |
| Neuroticism | | | | | | | |
| Not in isolation | 2.552 | .785 | 2.500 | .801 | $F(1,2104) = 3.030$, | $F(1,2104) = 30.896$, | $F(1,2104) = 6.186$, |
| In isolation | 2.802 | .772 | 2.811 | .754 | $p = .082$ | $p = .000$ | $p = .013$ |
| Extraversion | | | | | | | |
| Not in isolation | 3.126 | .649 | 3.152 | .656 | $F(1,2104) = 1.510$, | $F(1,2104) = 2.023$, | $F(1,2104) = 2.178$, |
| In isolation | 3.097 | .612 | 3.094 | .601 | $p = .219$ | $p = .155$ | $p = .140$ |
| Openness | | | | | | | |
| Not in isolation | 3.440 | .629 | 3.463 | .644 | $F(1,2104) = .000$, | $F(1,2104) = .900$, | $F(1,2104) = 5.402$, |
| In isolation | 3.498 | .610 | 3.474 | .622 | $p = .996$ | $p = .343$ | $p = .020$ |
| Agreeableness | | | | | | | |
| Not in isolation | 3.730 | .629 | 3.737 | .643 | $F(1,2104) = 1.738$, | $F(1,2104) = 31.487$, | $F(1,2104) = 4.091$, |
| In isolation | 3.579 | .648 | 3.544 | .650 | $p = .188$ | $p = .000$ | $p = .043$ |
| Conscientiousness | | | | | | | |
| Not in isolation | 3.923 | .689 | 3.935 | .711 | $F(1,2104) = 1.524$, | $F(1,2104) = 74.226$, | $F(1,2104) = 5.177$, |
| In isolation | 3.660 | .707 | 3.620 | .707 | $p = .217$ | $p = .000$ | $p = .023$ |
| Anxiety | | | | | | | |
| Not in isolation | 3.854 | .890 | 2.797 | .907 | $F(1,2104) = 6.805$, | $F(1,2104) = 37.463$, | $F(1,2104) = .644$, |
| In isolation | 3.096 | .887 | 3.067 | .855 | $p = .009$ | $p = .000$ | $p = .423$ |
| Depression | | | | | | | |
| Not in isolation | 2.375 | .891 | 2.308 | .890 | $F(1,2104) = .198$, | $F(1,2104) = 49.542$, | $F(1,2104) = 14.751$, |
| In isolation | 2.613 | .902 | 2.667 | .915 | $p = .656$ | $p = .000$ | $p = .000$ |
| Emotional Volatility | | | | | | | |
| Not in isolation | 2.426 | .870 | 2.396 | .877 | $F(1,2104) = .685$, | $F(1,2104) = 49.762$, | $F(2,104) = .970$, |
| In isolation | 2.696 | .844 | 2.699 | .856 | $p = .408$ | $p = .000$ | $p = .325$ |
| Sociability | | | | | | | |
| Not in isolation | 2.923 | .908 | 2.912 | .921 | $F(1,2104) = 1.473$, | $F(1,2104) = .086$, | $F(1,2104) = .039$, |
| In isolation | 2.916 | .877 | 2.896 | .842 | $p = .225$ | $p = .769$ | $p = .843$ |
| Assertiveness | | | | | | | |
| Not in isolation | 3.161 | .786 | 3.202 | .799 | $F(1,2104) = 5.436$, | $F(1,204) = .000$, | $F(1,2104) = .270$, |
| In isolation | 3.168 | .719 | 3.194 | .726 | $p = .020$ | $p = .983$ | $p = .603$ |
| Energy Level | | | | | | | |
| Not in isolation | 3.291 | .751 | 3.343 | .776 | $F(1,2104) = 1.743$, | $F(2,2204) = 11.315$, | $F(1,2204) = 4.579$, |
| In isolation | 3.205 | .779 | 3.193 | .760 | $p = .187$ | $p = .001$ | $p = .032$ |
| Curiosity | | | | | | | |
| Not in isolation | 3.542 | .738 | 3.568 | .734 | $F(1,2104) = .484$, | $F(1,2104) = .020$, | $F(1,2104) = 6.348$, |
| In isolation | 3.582 | .690 | 3.537 | .710 | $p = .487$ | $p = .887$ | $p = .012$ |
| Aesthetic Sensitivity | | | | | | | |
| Not in isolation | 3.248 | .843 | 3.272 | .857 | $F(1,2104) = .082$, | $F(1,2104) = 10.532$, | $F(1,2104) = 1.735$, |
| In isolation | 3.396 | .796 | 3.380 | .780 | $p = .775$ | $p = .001$ | $p = .188$ |
| Imagination | | | | | | | |
| Not in isolation | 3.530 | .766 | 3.550 | .778 | $F(1,2104) = .123$, | $F(1,2104) = .674$, | $F(1,2104) = .891$, |
| In isolation | 3.515 | .774 | 3.506 | .780 | $p = .725$ | $p = .412$ | $p = .345$ |
| Compassion | | | | | | | |
| Not in isolation | 3.810 | .760 | 3.819 | .762 | $F(1,2104) = .026$, | $F(1,2104) = 22.897$, | $F(1,2104) = .540$, |
| In isolation | 3.650 | .797 | 3.636 | .808 | $p = .872$ | $p = .000$ | $p = .463$ |
| Respectfulness | | | | | | | |

*(Continued)*

**Table 3.**  (Continued)

| Personality Trait | Pre | | Post | | Time | Isolation | Time x Isolation |
|---|---|---|---|---|---|---|---|
| | Mean | SD | Mean | SD | | | |
| Not in isolation | 4.036 | .735 | 4.022 | .748 | $F(1,2104) = 4.597,$ | $F(1,2104) = 36.918,$ | $F(1,2104) = 1.533,$ |
| In isolation | 3.836 | .820 | 3.788 | .829 | $p = .032$ | $p = .000$ | $p = .216$ |
| Trust | | | | | | | |
| Not in isolation | 3.343 | .742 | 3.368 | .760 | $F(1,2104) = .314,$ | $F(1,2104) = 12.810,$ | $F(1,2104) = 5.423,$ |
| In isolation | 3.251 | .761 | 3.210 | .710 | $p = .575$ | $p = .000$ | $p = .020$ |
| Organization | | | | | | | |
| Not in isolation | 3.942 | .814 | 3.939 | .826 | $F(1,2104) = 7.725,$ | $F(1,2104) = 45.271,$ | $F(1,2104) = 6.662,$ |
| In isolation | 3.721 | .848 | 3.864 | .843 | $p = .005$ | $p = .000$ | $p = .010$ |
| Productiveness | | | | | | | |
| Not in isolation | 3.870 | .787 | 3.906 | .791 | $F(1,2104) = 1.346,$ | $F(1,2104) = 77.163,$ | $F(1,2104) = 1.552,$ |
| In isolation | 3.562 | .819 | 3.561 | .819 | $p = .246$ | $p = .000$ | $p = .217$ |
| Responsibility | | | | | | | |
| Not in isolation | 3.956 | .743 | 3.960 | .760 | $F(1,2104) = 1.361,$ | $F(1,2104) = 61.517,$ | $F(1,2104) = 1.728,$ |
| In isolation | 3.696 | .803 | 3.661 | .776 | $p = .243$ | $p = .000$ | $p = .189$ |
| Responsibility[a] ^ | | | | | | | |
| Not in isolation | 4.056 | .734 | 4.064 | .762 | $F(1,2035) = .761,$ | $F(1,2035) = 72.456,$ | $F(1,2035) = 1.949,$ |
| In isolation | 3.777 | .810 | 3.740 | .780 | $p = .383$ | $p = .000$ | $p = .163$ |
| Dutifulness[a] | | | | | | | |
| Not in isolation | 3.950 | .598 | 3.906 | .587 | $F(1,1996) = 5.399,$ | $F(1,1996) = 53.849,$ | $F(1,1996) = 1.196,$ |
| In isolation | 3.728 | .623 | 3.712 | .627 | $p = .020$ | $p = .000$ | $p = .274$ |

$N$ = 2,106.

[a] Ns vary due to missing data.

^ From Roberts et al. [14].

large decrease on this item: Participants decreased in their willingness to go to work/school when sick, a decrease of nearly a one-half standard deviation. Further, the median item-total correlation fell from .207 at pretest to .089 at posttest, indicating that its relation with the other markers of Dutifulness changed. There was also evidence of change in three other items on the scale: Perceptions of not being dependable decreased, ignoring silly rules decreased, and following ethical principles increased (together these items show evidence of an increase in Dutifulness, as the first two items listed are reverse scored into the total). There was no change in the other four items (paying debts, following through on commitments, doing jobs carefully, and performing tasks conscientiously).

## Extraversion, openness, and agreeableness

The repeated measures ANOVA indicated that Extraversion increased slightly ($d$ = .03; Table 2). At the facet level, this increase was seen for Assertiveness and Energy Level but not Sociability. These changes were not significant when the covariates were included in the model (S1 Table). The increase in Energy Level occurred in participants who were not in isolation (Table 3). There were no other differences in change in the Extraversion domain or facets by age or gender (S2 and S3 Tables) or isolation status (Table 3). There was not a significant change in Openness ($d$ = .02) or Agreeableness ($d$ = .00) or any of their facets (Table 2) and change in these domains and facets was not moderated by age or gender (S2 and S3 Tables). There was, however, some moderation by isolation status (Table 3). Specifically, there was an

interaction for Openness, and specifically the facet of Curiosity, such that participants in isolation declined slightly in Openness and Curiosity whereas participants not in isolation increased slightly. A similar pattern emerged for the domain of Agreeableness and its facet of Trust. There were also baseline differences in Agreeableness by isolation status ($d$ = .24).

## Discussion

The present research suggests modest acute personality change during the initial stages of the coronavirus outbreak in the United States. Contrary to our hypothesis, there was a small decline in Neuroticism rather than the expected increase. This change in Neuroticism was only apparent among individuals who were not in quarantine/isolation. We likewise did not find the expected increase in Conscientiousness, and there was some evidence that the current social environment may have changed the meaning of an item. In exploratory analyses, there was modest evidence that isolation status moderated trait changes in Conscientiousness, Openness, and Agreeableness, as well as for Neuroticism.

Personality traits tend to be stable over time and resistant to normative life events that are stressful [8]. Of the five traits, there is the most evidence that Neuroticism may be the most reactive to stress. When individuals experience a great amount of distress, either through an extremely aversive event [17–19] or a depressive episode [9], Neuroticism tends to increase. A similar but weaker trend is found for long-term psychological responses to natural disasters, such as after the Christchurch Earthquake [20]. Likewise, interventions to improve mental health decrease Neuroticism [10]. Given the stress and anxiety over the coronavirus, we had expected Neuroticism to increase. Instead, the opposite pattern emerged. This decrease may be due to contrast effects. That is, reminders of the collective stress and anxiety that the world was under were everywhere: During the 10 days of the posttest data collection, there was significant volatility and losses in the stock market [21] (marker of economic anxiety), essential household products such as toilet paper were sold out across the country [22] (marker of consumer anxiety), and national polls indicated that 70% of American adults were concerned or very concerned about the virus in their community [23] (marker of individual anxiety). Feelings of personal stress and anxiety may be attributed less to the self when there is a tremendous amount of stress and anxiety experienced through the whole of society. In such a context, there might be an attenuated tendency to perceive and rate oneself as emotionally distressed as compared to other people. The stress and anxiety participants felt may have been ascribed to the external situation rather than their own personality. It is important to note that participants with pretest data but no posttest data scored higher in Neuroticism. This difference in attrition may have had an effect on the pattern of results. For example, as individuals higher in neuroticism were lost to follow-up, it is possible that this more emotionally vulnerable group responded differently to the pandemic. It is also of note, however, that the overall pattern that we found is consistent with anecdotal reports of decreases in anxiety among individuals who typically suffer from anxiety [24].

We did not find evidence for change in Conscientiousness. We hypothesized that the ubiquitous public health messaging to be more attentive to personal behavior would translate into an overall increase in a trait tendency to be conscientious, particularly the facet of Responsibility. Rather than Responsibility, however, we found only modest evidence for an increase in the facet of Productiveness, which indicated that individuals saw themselves as more efficient and persistent in this crisis. There was, however, a fascinating pattern for Dutifulness. Dutifulness measures the tendency to adhere strictly to ethical principles [15]. This trait tendency decreased between pre- and post-test, a change that primarily occurred in participants younger than 65 (i.e., working-aged adults). This decrease was due entirely to declines on one item

about going to work/school when not feeling well. In pre-pandemic times, this item was a fairly good marker of an individual's willingness to follow through on their commitments. The swift changes in the social landscape, however, may have changed the meaning of this item. Now, rather than a marker of conscientiousness, going to work/school while sick may be a marker of recklessness or antagonism, whereas staying at home and protecting one's community is conscientious. It is an example of how social context can (rapidly) change the meaning of an item and how it defines the trait it measures.

Approximately one-quarter of our sample reported being in isolation/quarantine within the last month. Our exploratory analysis suggested modest change in personality by isolation status. Of most note, isolation status moderated change in Neuroticism such that the decline in Neuroticism only occurred for those not in quarantine. Further, there was a cross-over interaction for the Depression facet: Individuals not in quarantine declined, whereas those in quarantine increased in a trait tendency toward depressed affect. Increases in depressed affect and other aspects of negative emotionality are common while in quarantine, and the effects may or may not be long lasting [25]. More generally, quarantine might provoke anxiety that is not assuaged by the stress and anxiety felt by the rest of the population. In addition to Neuroticism, isolation also moderated change in Openness, Agreeableness, and Conscientiousness. In all cases, these traits declined among individuals in isolation, specifically the facets of Curiosity, Trust, and Organization, respectively. The circumstances around isolation may lead to boredom and erode trust. There may also be less pressure to be organized because there is less that needs to get done in a timely manner. It is also of note that there were baseline differences in personality prior to quarantine. That is, individuals who go into quarantine had higher baseline levels of Neuroticism and lower Agreeableness and Conscientiousness. Individuals with these traits may be at greater risk of exposure through either who they interact with and/or they work jobs that put them at higher risk of exposure. Individuals higher in Neuroticism may also perceive more threat and go into quarantine to feel safer. There may also be bias associated with these traits in how quarantine/isolation is interpreted (e.g., safer at home may be interpreted as quarantine). We could not tease apart these different possibilities.

FFM personality traits are known to be stable [26] with normative changes across the lifespan [27] and are also known to be relatively resistant to change after normative life events [8]. As such, there would be no expectation that personality traits would change over just six weeks in normal circumstances. The coronavirus pandemic, however, is unprecedented in its disruption of daily life for most of the population. It was thus possible that it would also have an unprecedented effect on personality. As described above, extremely aversive and stressful events are associated with change in personality [17, 19], and the global scale of the current stressful event may have had a widely felt impact. And yet, even with the widespread fears over health consequences of complications of COVID-19, the economic uncertainty, and restrictions on daily life, personality traits have been mostly resistant to change. These findings support theoretical accounts of personality traits that argue for their stability [28], even in the face of acute environmental stressors. It may be the case that other aspects of psychological functioning, including state affect or mental health [29], may be more vulnerable to the impact of COVID-19 (but see [30]).

The present study had several strengths, including a pre-post design that captured trait psychological function just prior and during the acute phase of the coronavirus pandemic in the United States. The findings, however, need to be put in context. Although there was evidence of change, for example, the magnitude of change was small; in most cases, the change was less than one-tenth of a standard deviation. As such, overall there is more evidence of stability than substantial change. Still, personality would not be expected to change at all over such a short period of time in normal circumstances. The findings also need to be put in context of some

limitations. First, the attrition analysis indicated that there were significant selection effects for who remained in the sample at Time 2 that may have had an effect on the results, particularly for Neuroticism (as discussed above). It is important to note that this study was not originally designed to be longitudinal, so participants in the pretest survey did not know that they would be asked to complete a second survey. With the pandemic, the study was reconceptualized to take advantage of the data collected on psychological functioning just prior to the pandemic. Fortunately, many participants were willing to fill out a second survey, but given that the original study was not meant to be longitudinal, there was no expectation that participants would continue to participate. Second, we tested for trait change in the acute phase of the pandemic. Although the purpose of this measurement was to address whether trait psychological functioning was responsive to an acute health-threatening crisis, it is also possible that the effects of the crisis could take longer to consolidate into substantial changes in personality. Future work will need to address whether there are long-term changes in personality in response to the coronavirus pandemic. Future work also needs to address personality change during the pandemic in other cultural contexts. Third, our measure of quarantine/isolation was broad and did not differentiate between quarantine or isolation and the situation for the participant during quarantine/isolation (e.g., whether the person was alone or with another person). As such, we could not disentangle the exact circumstance of the quarantine/isolation and whether such differences are important for personality change. Finally, as with all non-experimental research, there may be other explanations for the current set of results that we cannot rule out.

Overall, the results suggest more trait psychological resilience than harm during the acute phase of the coronavirus spread and response in the United States. Consistent with the notion that traits are stable and resistant to change, there were few changes in response to the spread of the coronavirus and the measures to control the spread in the United States. The results further suggest that the broader social environment may be modifying both how individuals see themselves (e.g., attributing less anxiety and depressed affect to themselves) and the meaning of specific items to how they measure a trait (e.g., items of Dutifulness). Future work will need to address whether these modest changes are long lasting and/or whether different patterns of change emerge if this crisis is protracted.

## Supporting information

**S1 Fig. Flow chart of participants inclusion/exclusion.**
(DOCX)

**S1 Table. Mean change in personality traits between pretest and posttest controlling for covariates.** *N* = **2,137.** [a] Ns vary due to missing data. ^ From Roberts et al. [14].
(DOCX)

**S2 Table. Interaction between time and age on change in personality traits.** *N* = 2,137. [a] Ns vary due to missing data. ^ From Roberts et al. [14].
(DOCX)

**S3 Table. Interaction between time and gender on change in personality traits.** *N* = 2,137. [a]Ns vary due to missing data. ^ From Roberts et al. [14].
(DOCX)

**S4 Table. Mean change in dutifulness items between pretest and posttest.** *N*s range from 2,105 to 2,109 due to missing data. Items are not given in full because the NEO-PI-3 is protected by copywrite. [a] Reported in the raw metric but reverse scored into the total

Dutifulness score.
(DOCX)

## Author Contributions

**Conceptualization:** Angelina R. Sutin, Martina Luchetti, Damaris Aschwanden, Ji Hyun Lee, Amanda A. Sesker, Jason E. Strickhouser, Yannick Stephan, Antonio Terracciano.

**Data curation:** Angelina R. Sutin, Martina Luchetti, Antonio Terracciano.

**Formal analysis:** Angelina R. Sutin.

**Writing – original draft:** Angelina R. Sutin.

**Writing – review & editing:** Martina Luchetti, Damaris Aschwanden, Ji Hyun Lee, Amanda A. Sesker, Jason E. Strickhouser, Yannick Stephan, Antonio Terracciano.

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
