## [Decision Letter · Decision Letter 0]

22 Jun 2020

PONE-D-20-12661

Change in Five-Factor Model Personality Traits During the Acute Phase of the Coronavirus Pandemic

PLOS ONE

Dear Dr. Sutin,

Thank you for submitting your manuscript to PLOS ONE. After careful consideration, we feel that it has merit but does not fully meet PLOS ONE’s publication criteria as it currently stands. Therefore, we invite you to submit a revised version of the manuscript that addresses the points raised during the review process.

The manuscript has been evaluated by two reviewers. Please, see their comments appended at the bottom of this letter. As you will see, there were several major concerns that should be addressed in a reviewed version of your study.

We look forward to receiving your revised manuscript.

Kind regards,

Angel Blanch, Ph.D.

Academic Editor

PLOS ONE

Journal Requirements:

2. We note that you have stated that "Documentation of informed consent was waived because the data were collected and analyzed anonymously from web-based surveys". Please provide further information about the consent procedures for the study - please state what information participants were provided with before participation. Please clarify whether the participants provided online consent to take part in the study. If this was not received please state why not.

Reviewers' comments:

Reviewer's Responses to Questions

**Comments to the Author**

1. Is the manuscript technically sound, and do the data support the conclusions?

Reviewer #1: Yes

Reviewer #2: Partly

2. Has the statistical analysis been performed appropriately and rigorously? 

Reviewer #1: Yes

Reviewer #2: Yes

3. Have the authors made all data underlying the findings in their manuscript fully available?

Reviewer #1: Yes

Reviewer #2: Yes

4. Is the manuscript presented in an intelligible fashion and written in standard English?

Reviewer #1: Yes

Reviewer #2: Yes

5. Review Comments to the Author

Reviewer #1: I really appreciate this study, it provides information on the topic of stability and changes in personality traits during a stressful situation, specifically during the corona crisis. It is an important contribution to the knowledge of personality stability and change. I recommend the study for publication

I have only two comments:

1) Information that the quarantine / isolation has not been clearly defined and that it may have been interpreted differently should already be provided in the Method. There is a significant difference between quarantine from medical reasons and voluntary isolation. (“In the last month, I have been in quarantine / isolation because of the coronavirus.”)

2) The attrition of the sample between the first and the second measurement is substantial (approx. 45%). Moreover, participants who do not have posttest data differ from those who do have posttest data, not only in demographic variables but also in personality traits - they specifically have a higher level of neuroticism. Thus, the results might be significantly affected by who was willing to participate in the second measurement. I would appreciate it if the authors commented on this fact in more details in the discussion.

Reviewer #2: The authors try to test acute personality changes in response to the coronavirus pandemic using the Big Five Personality Model.

In my opinion, the sample size and the personality model used are the main strengths of the study. However, I think that the study has a series of limitations that make it difficult to respond to the initial objective. Furthermore, there are some aspects that should be reviewed and clarified.

- Some aspects of the procedure are not clear. I think it is necessary to include more information about the participants. Could people from all over the country participate? Was this information collected in the evaluation protocol? I think this variable can affect the results.

- Why include a band age 18-19? The others were 10 years old.

- There are other relevant variables that can influence the results. For example: employment situation, with whom the participants live, if they are sick or not, if they knew someone who had died of coronavirus or had the disease-

- I think that to understand the differences between people who were quarantined and those who were not, it is necessary to know the reasons why these people were quarantined: by their own decision? for being sick? by indications of the government? It is also important to know in what conditions they were in quarentena: alone or with family? Did they work or were they out of work? If the authors have not been able to collect this information, they should include it as an important limitation of the research.

- What information were participants given about the aims of the study when they were encouraged to participate in the second phase of the evaluation? What objective were they told the study had? In what context was the study presented in this phase? This information may also affect the results obtained.

- Were other evaluation instruments included in the second phase? And if so, in what order were this instruments presented, before or after the personality measure?

- I don't understand how some covariates were defined. Why include in the same group female and transgender/other/unknown? Was there any reason to divide the groups based on ethnicity and race? For example, where Asians were included?

- The authors indicate that exploratory analyzes examined whether there were larger changes in adults older than 65, because of the greater risk in this group. But it is also true that the older the fewer changes occur in personality traits. It is true that those over 65 are a more vulnerable group, but it may be that changes occur in other variables than traits (emotional states, worries ...). What do the authors think?

- I agree with the authors when they state that one of the limitations of the study is that it is likely that the effects of the crisis could take longer to consolidate into substantial changes in personality. This is an important limitation, considering that personality traits, by definition, are quite stable over time and, even more so, in adults. The authors also find that there is more stability than change. But the changes could occur in the long term. For this reason, the authors should clarify the significance of the results found.

In this context,the authors should make clear what the usefulness of this study is. What are its main practical or theorical implications? Is it useful at a clinical level, to identify coping strategies ...?

- I believe that the authors should make reference to the fact that there are many other explanations for the results and due to the limitations of the study, these other explanations cannot be ruled out.

In summary, I think the study is, but it is necessary to provide more information on the issues outlined above. Authors need to clarify some aspects of the methodology and that they highlight the limitations of the study.The authors should therefore be cautious in the conclusions of the study and clearly point it out.

It is necessary to clarify aspects of the method and procedure, it is necessary to clearly indicate the limitations, it is essential to better discuss the results.

6. PLOS authors have the option to publish the peer review history of their article (what does this mean?). If published, this will include your full peer review and any attached files.

Reviewer #1: No

Reviewer #2: No

---

## [Author Response · Author response to Decision Letter 0]

26 Jun 2020

Response to Reviewers

Journal Requirements:

We formatted the manuscript to be consistent with PLOS ONE’s style requirements.

2. We note that you have stated that "Documentation of informed consent was waived because the data were collected and analyzed anonymously from web-based surveys". Please provide further information about the consent procedures for the study - please state what information participants were provided with before participation. Please clarify whether the participants provided online consent to take part in the study. If this was not received please state why not.

We clarify that potential participants were given a description of the survey. Individuals clicked “yes” to indicate that they understand that they were being asked to complete a survey, that they were 18 years or older, and that they agreed to participate. Individuals who clicked yes were directed to the survey; those who clicked no were directed out of the study (pp. 4-5).

We formatted the Supporting Information in the manuscript as requested.

Reviewer #1

I really appreciate this study, it provides information on the topic of stability and changes in personality traits during a stressful situation, specifically during the corona crisis. It is an important contribution to the knowledge of personality stability and change. I recommend the study for publication

We thank the Reviewer for the overall positive evaluation of our manuscript.

I have only two comments:

1) Information that the quarantine / isolation has not been clearly defined and that it may have been interpreted differently should already be provided in the Method. There is a significant difference between quarantine from medical reasons and voluntary isolation. (“In the last month, I have been in quarantine / isolation because of the coronavirus.”)

We agree with the Reviewer and indicate this ambiguity in the Method (p. 8) as well as a limitation in the Discussion (pp. 22-23). We also dropped this result from the abstract to focus more on the primary findings.

2) The attrition of the sample between the first and the second measurement is substantial (approx. 45%). Moreover, participants who do not have posttest data differ from those who do have posttest data, not only in demographic variables but also in personality traits - they specifically have a higher level of neuroticism. Thus, the results might be significantly affected by who was willing to participate in the second measurement. I would appreciate it if the authors commented on this fact in more details in the discussion.

We now discuss the potential impact of attrition for the results for Neuroticism (p. 19) and discuss the issue of attrition more thoroughly in the limitations section in the Discussion (p. 22).

Reviewer #2

The authors try to test acute personality changes in response to the coronavirus pandemic using the Big Five Personality Model.

In my opinion, the sample size and the personality model used are the main strengths of the study. However, I think that the study has a series of limitations that make it difficult to respond to the initial objective. Furthermore, there are some aspects that should be reviewed and clarified.

We thank the Reviewer for the positive comments and suggestions for ways to improve the manuscript.

- Some aspects of the procedure are not clear. I think it is necessary to include more information about the participants. Could people from all over the country participate? Was this information collected in the evaluation protocol? I think this variable can affect the results.

Yes, participants were from all 50 states and Washington, DC and Puerto Rico. Sample sizes from each state/DC/PR were roughly proportionate to the population of each state/DC/PR. We now report this information in the Method (p. 5 and p. 6).

- Why include a band age 18-19? The others were 10 years old.

The age 18-19 age band was included to have representation of adolescents in the original sample. The range was limited to 18-19 to sample individuals who could consent legally for themselves. We report this information in the Method (p. 5).

- There are other relevant variables that can influence the results. For example: employment situation, with whom the participants live, if they are sick or not, if they knew someone who had died of coronavirus or had the disease-

Personality change was not moderated by employment (any type of employment versus not currently employed) or living situation (living alone versus living with at least one other person). At the time of the posttest assessment (mid-late March), relatively few people in the United States had tested positive for the coronavirus (about 100,000) or were known to have died from COVID-19 (about 2,000), so it is unlikely that enough participants in our sample were sick with coronavirus or knew someone who had died from it to be able to make meaningful comparisons. We now note in the limitations that there are other factors that may have contributed to the results (pp. 22-23).

- I think that to understand the differences between people who were quarantined and those who were not, it is necessary to know the reasons why these people were quarantined: by their own decision? for being sick? by indications of the government? It is also important to know in what conditions they were in quarentena: alone or with family? Did they work or were they out of work? If the authors have not been able to collect this information, they should include it as an important limitation of the research.

Unfortunately, we do not have this information. As stated in response to Reviewer 1, point #1 we now mention this missing information in both the Method (p. 8) and as a limitation in the Discussion (pp. 22-23).

- What information were participants given about the aims of the study when they were encouraged to participate in the second phase of the evaluation? What objective were they told the study had? In what context was the study presented in this phase? This information may also affect the results obtained.

The description of the survey was similar to the description of the first survey, except participants were told that they would also be asked questions about current events. Participants were also told that the survey would include some questions that were similar to what they might have been asked before and that it was important to answer the questions, even if they had answered them before. We added this information to the Method (p. 6).

- Were other evaluation instruments included in the second phase? And if so, in what order were this instruments presented, before or after the personality measure?

The second survey also included questionnaires about the coronavirus pandemic. The survey block with personality and the block with the coronavirus questionnaires were randomly counterbalanced across participants. There was no effect of order on the results for personality change. We now include this information in the Method (p. 6).

- I don't understand how some covariates were defined. Why include in the same group female and transgender/other/unknown? Was there any reason to divide the groups based on ethnicity and race? For example, where Asians were included?

A total of 22 participants (1.1%) indicated a gender identity other than male or female or chose not to report. The results were the same if this group was not included in the analysis. We added this information to the Method (p. 8). We included race (African American versus other) and ethnicity (Latinx versus other) as covariates because African Americans and Latinx populations have been disproportionately affected by the coronavirus and were the two largest racial/ethnic populations in our sample. Asians (n=168) were included in the other category (p. 9).

- The authors indicate that exploratory analyzes examined whether there were larger changes in adults older than 65, because of the greater risk in this group. But it is also true that the older the fewer changes occur in personality traits. It is true that those over 65 are a more vulnerable group, but it may be that changes occur in other variables than traits (emotional states, worries …). What do the authors think?

Yes, there is typically less change in personality in middle-age and older adulthood compared to younger populations. In normal circumstances, however, no change in personality would be expected in less than two months for any age group. We hypothesized change because of the extraordinary nature of the pandemic, and specifically for older adults because of the higher risk associated with age. Our focus was on potential change in traits, but, yes, the reviewer is correct that there could be changes in other aspects of psychological functioning, as we now note (p. 21).

- I agree with the authors when they state that one of the limitations of the study is that it is likely that the effects of the crisis could take longer to consolidate into substantial changes in personality. This is an important limitation, considering that personality traits, by definition, are quite stable over time and, even more so, in adults. The authors also find that there is more stability than change. But the changes could occur in the long term. For this reason, the authors should clarify the significance of the results found.

In this context,the authors should make clear what the usefulness of this study is. What are its main practical or theorical implications? Is it useful at a clinical level, to identify coping strategies ...?

There is some indication from the literature that aversive and stressful events are associated with personality change. And, although under normal circumstances there is no expectation that personality would change in a short period of time, the extraordinary nature of the pandemic may affect personality in ways that have not yet been observed. The present study had the opportunity to test this question with a strong design that measured personality just prior to the pandemic and again during a particularly stressful period early in the crisis in the United States. It is possible that there will be long-term changes in personality because of the pandemic, but the current research suggests that the initial stressful period in which routine daily life was substantially disrupted was not associated with large changes in personality. This research thus supports theoretical conceptualizations of FFM traits and adds that traits remain stable through such disruptions. We now address the theoretical implications of the study in greater depth in the Discussion (p. 21).

- I believe that the authors should make reference to the fact that there are many other explanations for the results and due to the limitations of the study, these other explanations cannot be ruled out.

We added this caveat to the limitations in the Discussion (pp. 22-23).

In summary, I think the study is, but it is necessary to provide more information on the issues outlined above. Authors need to clarify some aspects of the methodology and that they highlight the limitations of the study.The authors should therefore be cautious in the conclusions of the study and clearly point it out.

It is necessary to clarify aspects of the method and procedure, it is necessary to clearly indicate the limitations, it is essential to better discuss the results.

We have worked to address the points raised by the Reviewer to better clarify the context of our study and the limitations that are important to consider when evaluating the results.

---

## [Decision Letter · Decision Letter 1]

21 Jul 2020

Change in Five-Factor Model Personality Traits During the Acute Phase of the Coronavirus Pandemic

PONE-D-20-12661R1

Dear Dr. Sutin,

We’re pleased to inform you that your manuscript has been judged scientifically suitable for publication and will be formally accepted for publication once it meets all outstanding technical requirements.

Kind regards,

Angel Blanch, Ph.D.

Academic Editor

PLOS ONE

Additional Editor Comments (optional):

Reviewers' comments:

Reviewer's Responses to Questions

**Comments to the Author**

1. If the authors have adequately addressed your comments raised in a previous round of review and you feel that this manuscript is now acceptable for publication, you may indicate that here to bypass the “Comments to the Author” section, enter your conflict of interest statement in the “Confidential to Editor” section, and submit your "Accept" recommendation.

Reviewer #1: All comments have been addressed

Reviewer #2: All comments have been addressed

2. Is the manuscript technically sound, and do the data support the conclusions?

Reviewer #1: Yes

Reviewer #2: Yes

3. Has the statistical analysis been performed appropriately and rigorously? 

Reviewer #1: Yes

Reviewer #2: Yes

4. Have the authors made all data underlying the findings in their manuscript fully available?

Reviewer #1: Yes

Reviewer #2: Yes

5. Is the manuscript presented in an intelligible fashion and written in standard English?

Reviewer #1: Yes

Reviewer #2: Yes

6. Review Comments to the Author

Reviewer #1: (No Response)

Reviewer #2: I think the authors have included modifications that have improved the article. In my opinion, they have also incorporated information that clarifies some doubts in the methodology and results section. Although the limitations section could be extended, the authors have made an effort to incorporate the reviewers' suggestions. I also think that the main methodological difficulties of the study are included in the limitations section.

7. PLOS authors have the option to publish the peer review history of their article (what does this mean?). If published, this will include your full peer review and any attached files.

Reviewer #1: No

Reviewer #2: No

---

## [Editor Report · Acceptance letter]

28 Jul 2020

PONE-D-20-12661R1 

Change in Five-Factor Model Personality Traits During the Acute Phase of the Coronavirus Pandemic 

Dear Dr. Sutin:

I'm pleased to inform you that your manuscript has been deemed suitable for publication in PLOS ONE. Congratulations! Your manuscript is now with our production department. 

Kind regards, 

on behalf of

Dr. Angel Blanch 

Academic Editor

PLOS ONE